# Efficacy of Chemotherapy in Pain Control of Patients with Cancer at the Early Phase of Their Disease

**DOI:** 10.3390/healthcare13080931

**Published:** 2025-04-18

**Authors:** Ștefan Spînu, Daniel Sur, Veronica Creciun, Daniela Moșoiu, Daniel Ciurescu

**Affiliations:** 1Medlife Oncology Hospital, 500052 Brașov, Romania; stefan.spinu@medlife.ro; 2Medical Oncology Department, Oncology Institute “Prof. Dr. Ion Chiricuţă”, 400015 Cluj-Napoca, Romania; 3Department of Oncology, “Iuliu Hațieganu” University of Medicine and Pharmacy, 400347 Cluj-Napoca, Romania; 4Department of Hematology, Oncology Institute “Prof. Dr. Ion Chiricuţă”, 400015 Cluj-Napoca, Romania; creciun.veronica@elearn.umfcluj.ro; 5HOSPICE “Casa Speranței”, 500074 Brașov, Romania; daniela.mosoiu@hospice.ro; 6Departments of Medical and Surgical Specialties, Faculty of Medicine, Transylvania University of Brașov, 500019 Brașov, Romania; daniel.ciurescu@unitbv.ro; 7Medlife Hospital, 500152 Brașov, Romania

**Keywords:** palliative care, pain, chemotherapy, questionnaires

## Abstract

**Scope:** The rationale of the present study is to measure the impact of chemotherapy on the pain caused by the disease. **Materials and Methods:** The present study was based on the completion of two standardized questionnaires for the assessment of physical symptoms (Brief Pain Inventory—BPI—and the revised Edmonton Symptom Assessment System—ESASr) in three different moments. The research was conducted at the Oncology Institute “Prof. Dr. Ion Chiricuță” Cluj-Napoca between 2022 and 2023. **Results:** 24 patients were included in the study, and they received different cytostatic treatment regimens, adapted to the underlying pathology. We analyzed the distribution of all symptoms from the ESASr and the total score. In terms of the general situation, if we exclude pain, there is never a difference between Time 2 and 3. Furthermore, in all cases where there is a statistically significant effect, there is a difference between Time 1 and 3. We also analyzed the distribution of activities with which pain interacts in BPI. Regarding the general situation, three patterns of influence of pain on the examined parameters are found: for general activity, mood, ability to work, and sleep, there are significant differences between moments 1 and 2 and 1 and 3, without significant differences between moments 2 and 3. The second pattern identified refers to the lack of effect of pain on the ability to walk and on the relationship with other people. The peculiarity is represented by the influence on the enjoyment of life that was significantly improved only from Time 1 to 3. **Conclusions:** The present study demonstrated a considerable effect of cytostatic treatment on the management of cancer-related pain, as shown across different evaluations.

## 1. Introduction

Palliative care is a strategic approach whose main goal is to increase the quality of life of patients. For this, it is necessary that a multidisciplinary team collaborates so patients and their families control the symptoms associated with the underlying disease, but also benefit from psychosocial and spiritual support.

Pain has had a unanimously accepted definition since 1979, namely that it is an unpleasant sensory and emotional experience associated with injuries [1]. In this context, pain is one of the most worrying symptoms associated with cancer, but also one of the most difficult charges to manage. This happens especially in the absence of a thorough understanding of the pathophysiological mechanisms underlying it.

According to published research, multimodal cancer therapies may be able to reduce the pain brought on by the cancer itself and, moreover, its intensity also takes into account the other dimensions of total pain (psychic, social, and spiritual components). However, a number of cytotoxic substances have the potential to cause neurotoxicity over time, which can lead to neuropathic pain [2,3,4].

The rationale of the present study is to measure the impact of antineoplastic treatment (chemotherapy) on the pain caused by the disease.

## 2. Materials and Methods

### Study Design

The present study was analytical, observational, quantitative, longitudinal, prospective, and non-randomized. It was based on the completion of two standardized questionnaires for the assessment of physical symptoms (Brief Pain Inventory—BPI—and revised Edmonton Symptom Assessment System—ESASr) in three different moments, selected exclusively based on chemotherapy cycles (every two and three weeks, respectively) depending on the chemotherapy regimen used for the respective pathology. This study aims to identify how the administration of chemotherapy can influence the management of cancer-related pain.

The research was carried out in the Medical Oncology section (including both outpatient, day hospital, and continuous hospitalization) of the Oncology Institute “Prof. Dr. Ion Chiricuță” Cluj-Napoca. The duration of the study was between the years 2022 and 2023.

Inclusion and exclusion criteria:

*Inclusion criteria*: patients over 18 years, recently diagnosed with cancer and pain syndrome and who will undergo oncology treatment.

*Exclusion criteria*:-Patients who refuse to participate in the study after being informed about it;-Uncooperative patients/with altered cognitive status;-Patients with pain syndrome caused by other pathologies other than cancer;-Patients with at least two types of cancer;-Patients who have a history of radio- and/or chemotherapy for another neoplasia;-Patients with a life expectancy lower than the established follow-up period.

Limitations of the study: the small number of patients, lack of a control arm, single institutional study, heterogeneous group of patients with different malignancies, and chemotherapeutic treatments were included in the study.

Ethical committee and informed consent:

The patients who participated in the study were informed about how it was conducted and expressed their written consent regarding the acceptance of participation. The initiation of the project took place after the opinion of the Ethics Commission of the Oncology Institute “Prof. Dr. Ion Chiricuță” Cluj-Napoca, having a favorable decision to conduct study no. 1360/2 February 2022, with evaluation report no. 233/3 February 2022. Each patient who met the study’s inclusion and exclusion criteria received a preliminary meeting from the principal investigator. In this way, the patient was informed about the purpose of the study, the applied methods (with an emphasis on the use of a technical medical language adapted to the level of training and understanding of the patient), the methods/purposes of the eventual use of the obtained medical data, the policy of the study related to the use of the personal data. Regarding the method of completing the questionnaires, at the first assessment, 92% needed help, while at the 2nd assessment, only 67% were helped, and at the last assessment, the percentage of those helped in completing it was 58%.

Patient data collection and clinical characteristics:

After analyzing the collected data, 24 patients were included in the study, of which 18 were men, representing 75% of the total.

The patients’ ages ranged from 42 to 75 years, with a mean of 59 years. The patients were assigned to the Oncology Institute “Prof. Dr. Ion Chiricuță” to different doctors (6 medical oncologists in total) and treated for various neoplastic conditions (head and neck cancer—12 patients; gastric cancer—3 patients; sarcoma and lung cancer—2 patients each; and 1 patient each with breast cancer, colon cancer, penile cancer, bladder cancer, and ovarian cancer) starting from February 2022 and until March 2023, both in day hospitalization and in continuous hospitalization.

Treatment schedules and number of chemotherapy cycles

The patients received different cytostatic treatment regimens, adapted to the underlying pathology, as follows: one of the patients had a treatment regimen consisting of a single cytostatic, 6 patients received a combination of two therapeutic agents, and the vast majority (17 patients) were subjected to a regimen of 3 chemotherapy drugs. The main chemotherapy association consists of platinum compounds (cisplatin, carboplatin, oxaliplatin), antimetabolits (5-fluorouracil, capecitabine), anthracyclines (doxorubicin, epirubicin), vinca alkaloids (vincristine), and alkylating agents (ifosfamide) and antimicrotubule agents (paclitaxel). All the aforementioned chemotherapy drugs were adapted to the patients’ clinical and biological statuses.

Statistical analysis:

According to data distribution, for the correlation of the number of cytostatic drugs used for the patient’s treatment and improvement in average pain intensity/24, the Shapiro–Wilk test was applied. Also, the data arrangement from the questionnaires in the form of “Boxplot” graphs displays the distribution of a continuous variable.

## 3. Results

To check whether the reduction in pain is significant, we first analyzed the distribution of all symptoms from the ESASr and the total score (Appendix A).

In terms of the general situation, if we exclude pain, there is never a difference between Time 2 and 3. Furthermore, in all cases where there is a statistically significant effect, there is a difference between Time 1 and 3.

The arrangement of data from the ESASr questionnaire in the form of “Boxplot” graphs displays the distribution of a continuous variable (Appendix A).

Similar to the analysis of the distribution of data from the ESASr questionnaire, we analyzed the distribution of activities with which pain interacts in BPI (Appendix A).

Regarding the general situation, three patterns of influence of pain on the examined parameters are found: for general activity, mood, ability to work, and sleep, there are significant differences between moments 1 and 2 and 1 and 3, without significant differences between moments 2 and 3. The second pattern identified refers to the lack of effect of pain on the ability to walk and on the relationship with other people. The peculiarity is represented by the influence on the enjoyment of life that was significantly improved only from Time 1 to 3, not between 1 and 2 or 2 and 3.

The following figures in the Appendix A show the distribution of continuous variables in a manner identical to that previously reported for the ESASr questionnaire (in the form of Boxplot graphs).

Regarding the average pain intensity parameter, it is found that at assessment 1, its average score is 4 on the visual analogue scale. Following that, at assessment 2, it decreases to 2.20 on the VAS (Visual Analogue Scale), and in the last assessment, it reaches 1.33. All reductions were statistically significant across examinations. Analyzing the percentage of pain relief (as reported by the patient) by the prescribed painkiller treatment, it was revealed that at assessment 1, 21 of the 24 patients were receiving a pain relief preparation, with an average pain reduction of 80%. For evaluation number 2, only 14 of the 24 patients were still receiving pain treatment, and in this case, the average pain relief was 71%. For the last assessment, 9 out of the 24 patients were still receiving pain medication, and for them, the relief was, on average, 86%.

Table 1 was created to highlight which level of pain relief treatment (according to the WHO scale) the patients used for each of the evaluations.

It is therefore observed in Table 1 that the biggest difference is recorded in the “no treatment” arm, where the percentage of patients increased considerably from one examination to another (13%–42%–63%). Also, for those who opted for first-step painkillers, their number decreased statistically significantly over time (50%–13%–0%). Conversely, for the groups of patients treated with opioids (steps 2 and 3), the percentages of patients remained relatively constant throughout the study, as follows: for step 2 (33%–37%–29%), and for step 3 (4%–8%–8%).

When dose reduction or discontinuation of pain therapy at assessment two (versus first) was considered, this was observed to occur in 46% of cases. Moreover, it was desired to observe the phenomenon and compare evaluation three with two, where the reduction/stopping occurred additionally in 42% of the cases.

In order to establish a correlation between the number of cytostatics used and the improvement in average pain intensity/24 h, the Shapiro–Wilk test was initially used to analyze the distribution of the data (Table 2).

The Shapiro–Wilk test reveals that the average pain intensity/24 h variable is not normally distributed. This time, given the non-normality of the data and the mixture of a between-subject effect (number of cytostatic drugs used) and a within-subject effect (time), we opted for a Generalized Linear Model with gamma distribution (ideal for heavily skewed data) and inverse link function.

Since from the group of patients, only one patient received only one chemotherapeutic agent, we performed the analysis after excluding him.

It is observed that time has a statistically significant effect on the reduction of average pain intensity/24 h (*p* < 0.001), but for the number of antitumor agents used or for the interaction between it and time, the situation remains unchanged (no effect) (Table 3).

The models that we mentioned earlier tell us that, despite the fact that time leads to a reduction in average pain intensity, the number of antitumor agents used has no impact on the improvement of this symptom. The behaviors of the three classes of participants (classes divided by the number of chemotherapeutic agents) are very similar (Figure 1).

Figure 2 compares the average pain intensity/24 h both for patients for whom the dose of the analgesic was reduced/stopped and for those who continued with the same dose on the occasion of evaluations 2 and 3.

It is, therefore, found that at the second assessment, those who did not reduce or stop the painkiller treatment presented a higher average pain intensity/24 h compared to those whose therapeutic plan against pain was adjusted (in the sense of its de-escalation). In evaluation number 3, the pain was felt more intensely by those who benefited from the reduction of analgesic doses, the opposite situation to the one encountered at the time of the second evaluation.

## 4. Discussion

In order to be able to analyze the situation of the effectiveness of oncology treatments in pain control in patients with neoplastic disease at diagnosis within the Oncology Institute “Prof. Dr. Ion Chiricuță” Cluj-Napoca, a comprehensive analysis of the results of the studies present in the literature is necessary.

A 40-year systematic review of more than 50 clinical trials, published in 2007, shows that the prevalence of pain in cancer patients is different according to four subgroups: in those who have completed their treatment, this was found in 33% of cases. Among those undergoing therapy, the prevalence was 59%. In terminally ill patients with advanced or metastatic disease, pain was reported in 64% of cases. When analyzing all patients, regardless of disease stage, the overall prevalence was 53%. More than a third of the patients specified that the intensity of the pain was moderate or severe, and the tumor location with the largest number of patients suffering from pain was that of the ENT [5].

Since 1986, the question has been raised whether the administration of chemotherapy intra-arterially, selectively, can relieve pain in people with unresectable recurrence of rectal cancer. In research published in the American Journal of Surgery, the authors studied a group of 10 patients who were continuously infused with 5-fluorouracil in bilateral iliac arteries for 7 days at a dose of 800 mg/m^2^. Also, at the end of these 7 days, an additional bolus injection of mitomycin C was performed at a dose equal to 10 mg/m^2^. Furthermore, 4 of the 10 patients also benefited from hyperthermia with a microwave generator on the second and fifth day of continuous chemotherapy infusion. Three of six patients who received intra-arterial chemotherapy alone experienced significant pain relief, and all four who received additional hyperthermia achieved pain control [6].

ASTRO published a study in 2021 comparing the quality of life of patients with locally advanced rectal cancer according to the treatment received. One group was treated with chemoradiotherapy and a boost dose of radiation to the tumor bed (5 fractions of 3 Gy each) compared to the control group that received standard chemoradiotherapy. Patients were randomized into the two groups and subsequently followed for 2 years. Those who received escalated doses of chemoradiotherapy (the additional dose boost group) experienced a significant deterioration in quality of life during the first year of follow-up. This translated into more local pain, more fatigue and diarrhea, and a decrease in the person’s social functioning. From the second year of follow-up, the two groups became homogeneous in terms of globally assessed quality of life [7].

A case published in the Japanese literature shows that an Asian-only fluorouracil derivative, S-1 (tegafur/gimeracil/oteracil), used as second-line treatment after gemcitabine in a patient with unresectable pancreatic neoplasia significantly improved the patient’s abdominal pain complaints [8].

Given that bone metastases are common in breast cancer patients, and the vast majority of these secondary findings cause pain, it is important that their approach be as effective as possible. That is why Shimaa Ahmed led a study that directly compared chemotherapy (with capecitabine) together with external irradiation (group A) versus irradiation alone (group B). The mean pain score decreased for both groups from the first week of treatment, and at week 4, a significant decrease in this score was noted. The difference between the mean scores of the two study groups was statistically significant, with the lowest value of *p* (0.032) at week twelve. Twenty-eight days after the initiation of treatment, 42.9% of the subjects in Group A reported no pain, compared to only 19% in Group B. The conclusion of the study was that the addition of chemotherapy to radiotherapy is superior to radiotherapy alone in controlling the pain of bone metastases from breast cancer [9].

## 5. Conclusions

Nowadays there is a rising awareness of the importance of managing cancer symptoms, particularly pain. This is due to the fact that neoplasias are still a major public health concern worldwide, with rising incidence and mortality rates expected in the upcoming years. Since it can refer to a wide range of illnesses with similar features, including potential involvement of all organs and age of beginning, cancer is truly an umbrella term.

The present study demonstrated a considerable effect of cytostatic treatment on the management of cancer-related pain, as shown across different evaluations. Moreover, the ESASr total score also improved during follow-up, with statistically significant differences between Times 1 and 2 and 1 and 3, respectively. Except for pain, no symptoms improved significantly between examinations 2 and 3. For the “Brief Pain Inventory” questionnaire, for the ability to walk and for the influence on relationships with other people, chemotherapy did not prove its effectiveness; instead, the enjoyment of life was significantly improved after two cycles of treatment, and for the other symptoms (general activity, mood, ability to work and sleep), chemotherapeutic agents achieved statistically significant improvements between Times 1 and 2 and 1 and 3, without any influence on the Time 2 and 3 difference.

During the follow-up, it was highlighted that the number of patients who remained without analgesic treatment increased in an important way from one examination to another, and this was mainly due to those who were stopped from the step one treatment.

The number of cytostatics did not matter in the equation of the reduction in average pain intensity, with the most important parameter involved being the time during which the cytostatic treatment regimen worked.

Studies with a similar design were not found as such. However, there has been research that has proven the effectiveness of administering antineoplastic therapies (systemic and/or local) to improve all aspects related to quality of life.

## Figures and Tables

**Figure 1 healthcare-13-00931-f001:**
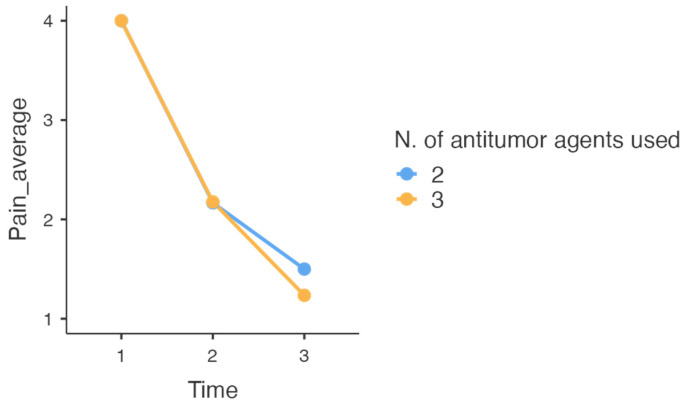
The relationship between time, the number of cytostatics used (two or three), and the average pain intensity.

**Figure 2 healthcare-13-00931-f002:**
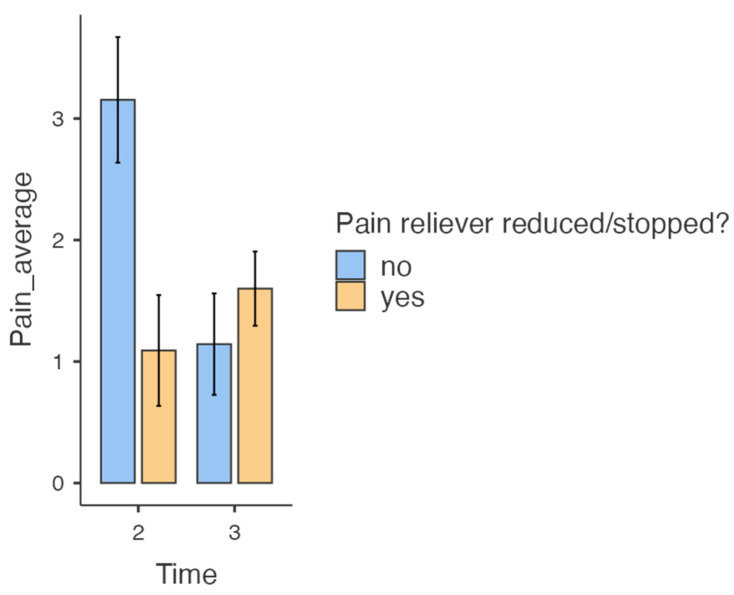
The ratio between the average intensity of pain/24 h and the reduction/stopping or not of the analgesic.

**Table 1 healthcare-13-00931-t001:** The relationship between the step of pain treatment used and the percentage of patients for each of the 3 evaluations.

Evaluation Number	Step of Pain Management	Percentage of Patients (%)
**1**	without treatment	13
	1	50
	2	33
	3	4
**2**	without treatment	42
	1	13
	2	37
	3	8
**3**	without treatment	63
	1	0
	2	29
	3	8

**Table 2 healthcare-13-00931-t002:** Distribution of data on the average pain intensity parameter.

				Skewness		Kurtosis		Shapiro–Wilk	
	Mean	Median	SD	Skewness	SE	Kurtosis	SE	W	*p*
Pain_average	2.514	2	2.162	0.93	0.283	0.593	0.559	0.904	<0.001

**Table 3 healthcare-13-00931-t003:** The involvement of the number of cytostatics used (two or three) and time in the improvement of average pain intensity/24 h.

	X²	df	*p*
N. of antitumor agents used	0.141	1	0.707
Time	13.882	2	<0 .001
N. of antitumor agents used ✻ Time	0.204	2	0.903

## Data Availability

No new data were created or analyzed in this study. Data sharing does not apply to this article.

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
