# Peer review of "Efficacy of Chemotherapy in Pain Control of Patients with Cancer at the Early Phase of Their Disease"

_healthcare, 2025, doi:10.3390/healthcare13080931_

Round 1

Reviewer 1 Report

Comments and Suggestions for Authors

This study is to measure the impact of chemotherapy on the pain caused by the disease. However,because the different kinds of cancer have  different sensitivity to chemotherapy. It is hard to measure the time of pain relieved by chemotherapy.  

The author should explain the reason that the significant differences between  moments 1-2 and 1-3, and without significant differences between moments 2 -3.(line 29.30.31 )and what kind of statistic method to be used in this part.

Author Response

Article healthcare-3506356

Title: “Efficacy of chemotherapy in pain control of patients with cancer at the early phase of their disease” by Spinu et al.

Reviewer’s response

Reviewer 1

We want to thank Reviewer 1 for the time spent evaluating our article.The suggestions of the reviewer are on point and by responding to them we can improve our article.

1 - This study is to measure the impact of chemotherapy on the pain caused by the disease. However,because the different kinds of cancer have  different sensitivity to chemotherapy. It is hard to measure the time of pain relieved by chemotherapy. 

Response: We agree that each neoplasm has a different sensitivity to chemotherapy and that it is difficult to measure the time to pain relief with chemotherapy in a general way. However, this pilot study aimed to open the way to new research opportunities that would answer more precise questions about what happens to pain depending on specific tumor locations and specific cytostatics used.

2- The author should explain the reason that the significant differences between  moments 1-2 and 1-3, and without significant differences between moments 2 -3.(line 29.30.31 )and what kind of statistic method to be used in this part.

Response: Between times 2 and 3 there were not always statistically significant differences due to the fact that the best response to treatment was recorded most impressively after the first cycle of chemotherapy, then the decrease in the intensity of symptoms was smaller.

Reviewer 2 Report

Comments and Suggestions for Authors

1- Different cancers have varying pain profiles and responses to chemotherapy. Without specifying the cancer type(s), the results cannot be meaningfully interpreted or applied to clinical settings.

2- The study does not differentiate between tumor-related pain and chemotherapy-induced neuropathy. These are distinct pain mechanisms with different treatment responses.

3- The sample size is too small to draw statistically meaningful conclusions. Given inter-individual variability in pain perception and response to treatment, a larger cohort is needed for robust analysis.

4-Without a control group (e.g., patients not receiving chemotherapy), it is impossible to determine whether chemotherapy directly affects pain levels or if changes are due to other factors such as supportive care or disease progression.

5- The study measures pain at three different time points but does not justify their selection. Are these based on chemotherapy cycles, disease progression, or another criterion? A rationale is necessary.

6-  Pain assessment using only BPI and ESASr is subjective. The inclusion of objective measures (e.g., analgesic consumption, biomarkers of inflammation) would improve the study's reliability.

7-  It is unclear whether chemotherapy directly reduces pain or if improvements are due to factors such as analgesic use, psychological adaptation, or tumor shrinkage.

Author Response

Article healthcare-3506356

Title: “Efficacy of chemotherapy in pain control of patients with cancer at the early phase of their disease” by Spinu et al.

Reviewer’s response

Reviewer 2

We want to thank Reviewer 2 for the time spent evaluating our article. The suggestions and observations of Reviewer 2 can help us improve our article.

1- Different cancers have varying pain profiles and responses to chemotherapy. Without specifying the cancer type(s), the results cannot be meaningfully interpreted or applied to clinical settings.

Response: We agree that each neoplasm has a different sensitivity to chemotherapy and that it is difficult to measure the time to pain relief with chemotherapy in a general way. However, this pilot study aimed to open the way to new research opportunities that would answer more precise questions about what happens to pain depending on specific tumor locations and specific cytostatics used.

2- The study does not differentiate between tumor-related pain and chemotherapy-induced neuropathy. These are distinct pain mechanisms with different treatment responses.

Response: In our study, we focused on selecting patients who did not have pre-existing neuropathy and who presented with pain syndrome generated by the cancer itself. As we have already shown in the study, this type of pain was significantly improved from one examination to another, and neurotoxicity did not set in in any of the subjects .

3- The sample size is too small to draw statistically meaningful conclusions. Given inter-individual variability in pain perception and response to treatment, a larger cohort is needed for robust analysis.

Response: We agree that the patient cohort was limited and that for these reasons the statistical power of the study is low. However, the study we conducted remains one that will open new possibilities for further investigation in more specific situations.

4-Without a control group (e.g., patients not receiving chemotherapy), it is impossible to determine whether chemotherapy directly affects pain levels or if changes are due to other factors such as supportive care or disease progression.

Response: We agree that we did not monitor the results by comparison with a control group, but we did not consider it ethical and necessary considering that ideally - this control group should have been represented by patients who did not receive systemic treatment, but only anti-pain treatment. The pain relief and reduction in the need for pain treatment in our patients occurred through the antitumor response of chemotherapy.

5- The study measures pain at three different time points but does not justify their selection. Are these based on chemotherapy cycles, disease progression, or another criterion? A rationale is necessary.

Response: The three times were selected exclusively based on chemotherapy cycles (every two and three weeks, respectively)

6-  Pain assessment using only BPI and ESASr is subjective. The inclusion of objective measures (e.g., analgesic consumption, biomarkers of inflammation) would improve the study's reliability.

Response: In preparing a more complex study, we will also consider objective parameters (including biomarkers).

7-  It is unclear whether chemotherapy directly reduces pain or if improvements are due to factors such as analgesic use, psychological adaptation, or tumor shrinkage.

Response: The pain relief and reduction in the need for pain treatment in our patients occurred through the antitumor response of chemotherapy, most likely.

Reviewer 3 Report

Comments and Suggestions for Authors

Page 1: Abstract: Line 26, 27: Please define what times 1, 2 and 3 are. What was the time interval between the 3 times?

Line 33: Please also describe what kind of pain interventions were performed between times 1 through 3. Could other pain interventions (Opioids, NSAIDs, radiation, surgical/interventional radiologic interventions) be contributing to the improvement in pain scores? It is important to acknowledge potential confounders.

Page 2: line 60: I would recommend using a different phrase instead of "three different moments". Try "at three different points of time" instead. I also request the authors to elaborate on what these three different time points were. Were they at equal time intervals for all patients, or based on how many treatment cycles for the underlying malignancy were delivered, or were they arbitrary?

Line 74: Please rewrite this exclusion criterion to be more precise: Did the authors mean patients with more than one malignancy? Or specifically just patients with two types of cancer were excluded, and patients with three types of cancers could be included?

Line 78: I would replace "Ethical" with "Ethics". 

Line 89: did the authors mean to write "personal data"?

Page 3: Line 98: "at the Oncological Institute"

Line 113:  "were adapted to the patients' clinical and biological status". Add apostrophe to patients. 

Page 4: Line 146: please define VAS. 

Table 1: I would re-label the third column of the table as Percentage of patients instead of number of patients, unless the authors intend to provide the number of patients in addition to the percentages. 

Page 6: Line 207 - 210: Please rewrite these sentences for clarity and ease of understanding. In the current form, they are cumbersome to read and difficult to understand. 

Line 215-223: I am not sure how the description of this study is relevant to the analysis being reported in this manuscript. I would recommend removing this paragraph. I don't think any of the patients in this study received intra-arterial chemotherapy, unless the authors have not mentioned it in the manuscript. 

Page 7: Line 232: more "local" pain

Line 233: What do the authors mean by "person's social input"? Please clarify. 

Line 245-246: I would not use the phrase "p value being minimal". 

I can appreciate the effort that the authors have put into conducting this study and doing a comprehensive analysis. However, I do have some significant concerns regarding the premise of the study and the methodology that was employed. 

The study has included a rather diverse group of cancer patients. The amount of pain that the patients would be expected to have prior to starting treatment would be highly dependent on the disease site. Most patients with cancer pain would be expected to have improvement with treatment, and thus the claim that they had improvement with chemotherapy is not a novel or radical claim, it is to be expected. The extent of improvement observed is less likely to be affected by the number of different cytotoxic chemotherapeutic agents used to treat the different cancers. Again, it would be inappropriate to use number of chemotherapeutic agents used as a variable to assess pain relief, as all different cancers have different chemotherapy regimens that are considered standard of care, and the number of agents used differ among different cancers. 

The patients that required step 2 and 3 pain management remained relatively stable. This makes me think that there was a specific group of patients (my guess would be head and neck cancer patients), who are particularly more vulnerable to pain, due to the modality of treatment employed (chemoradiation) which increases their pain as the treatment goes on, and pain improves only several weeks after the treatment is completed. 

Furthermore, the authors have not specified what the time intervals were between the moments 1, 2 and 3 in the pain evaluations. 

Due to these concerns, I feel that the manuscript still needs significant changes before consideration for publication. 

Potential suggestions for changes: include the 12 head and neck cancer patients and exclude the other patients. This will help compare scores and improvements in a uniform and homogenous population, making the results more meaningful. Exclude the patients with other cancers as their location and pathophysiology and treatment approaches are significantly different and an N of 3, 2 or 1 does not add much to the overall study. 

I suggest the authors to address if other palliative approaches such as radiation were utilized for pain relief in these patients. 

I would recommend that the authors address the limitations of the study, including the small number of patients, lack of a control arm, single institutional study, etc. 

Comments on the Quality of English Language

There is room for the language in the manuscript to be more precise and professional. The manuscript could also be re-written to make it easier to read and understand. There are instances in the manuscript where better word choice is possible, and the statements can be phrased better. 

Author Response

Article healthcare-3506356

Title: “Efficacy of chemotherapy in pain control of patients with cancer at the early phase of their disease” by Spinu et al.

Reviewer’s response

Reviewer 3

We want to thank Reviewer 3 for the effort in analyzing our article. We consider that by answering the requests of the reviewer we can make the article more easy to read for the Healthcare Journal readership. 

Page 1: Abstract: Line 26, 27: Please define what times 1, 2 and 3 are. What was the time interval between the 3 times?

Response:The three times were selected exclusively based on chemotherapy cycles (every two and three weeks, respectively)

Line 33: Please also describe what kind of pain interventions were performed between times 1 through 3. Could other pain interventions (Opioids, NSAIDs, radiation, surgical/interventional radiologic interventions) be contributing to the improvement in pain scores? It is important to acknowledge potential confounders.

Response:For the patients in the study, no invasive methods of pain reduction (surgery or radiotherapy) were used, but exclusively analgesic treatment (all stages of pain treatment were used)

Page 2: line 60: I would recommend using a different phrase instead of "three different moments". Try "at three different points of time" instead. I also request the authors to elaborate on what these three different time points were. Were they at equal time intervals for all patients, or based on how many treatment cycles for the underlying malignancy were delivered, or were they arbitrary?

Response: The three times were selected exclusively based on chemotherapy cycles (every two and three weeks, respectively) depending on the chemotherapy regimen used for the respective pathology.

Line 74: Please rewrite this exclusion criterion to be more precise: Did the authors mean patients with more than one malignancy? Or specifically just patients with two types of cancer were excluded, and patients with three types of cancers could be included?

Response: All patients with at least two neoplasms were excluded.

Line 78: I would replace "Ethical" with "Ethics".

Response:Agree with the change.We modified the text accordingly.

Line 89: did the authors mean to write "personal data"?

Personal data is the correct form.

Page 3: Line 98: "at the Oncological Institute"

Respose: Oncology Institute is the right choice . We amended the text.

Line 113:  "were adapted to the patients' clinical and biological status". Add apostrophe to patients.

Response: We added an apostrophe in the indicated place.

Page 4: Line 146: please define VAS.

Response: Thank you for your suggestion.We have defined VAS: Visual Analogue Scale.

Table 1: I would re-label the third column of the table as Percentage of patients instead of number of patients, unless the authors intend to provide the number of patients in addition to the percentages.

Response: We redefined the last column according to the instructions.

Page 6: Line 207 - 210: Please rewrite these sentences for clarity and ease of understanding. In the current form, they are cumbersome to read and difficult to understand.

Response: We have replaced the text for clarity: ”This was found in 33% of cases. Among those undergoing therapy, the prevalence was 59%. In terminally ill patients with advanced or metastatic disease, pain was reported in 64% of cases. When analyzing all patients, regardless of disease stage, the overall prevalence was 53%”.

Line 215-223: I am not sure how the description of this study is relevant to the analysis being reported in this manuscript. I would recommend removing this paragraph. I don't think any of the patients in this study received intra-arterial chemotherapy, unless the authors have not mentioned it in the manuscript.

Response: Indeed, no patient received intra-arterial chemotherapy in our study, so we chose to cite this article to make a comparison of how another way of administering cytostatic treatment (much more targeted) can lead to pain relief.

Page 7: Line 232: more "local" pain

Response: We have corrected

Line 233: What do the authors mean by "person's social input"? Please clarify.

Response: We replaced it with "social functioning"

Line 245-246: I would not use the phrase "p value being minimal".

Response: We replaced it with ” With the lowest value of p”

I can appreciate the effort that the authors have put into conducting this study and doing a comprehensive analysis. However, I do have some significant concerns regarding the premise of the study and the methodology that was employed.

The study has included a rather diverse group of cancer patients. The amount of pain that the patients would be expected to have prior to starting treatment would be highly dependent on the disease site. Most patients with cancer pain would be expected to have improvement with treatment, and thus the claim that they had improvement with chemotherapy is not a novel or radical claim, it is to be expected. The extent of improvement observed is less likely to be affected by the number of different cytotoxic chemotherapeutic agents used to treat the different cancers. Again, it would be inappropriate to use number of chemotherapeutic agents used as a variable to assess pain relief, as all different cancers have different chemotherapy regimens that are considered standard of care, and the number of agents used differ among different cancers.

Response:We considered the number of chemotherapeutic agents thinking that the more aggressive we are towards the tumor, the greater the response in terms of decreasing pain intensity could be.

The patients that required step 2 and 3 pain management remained relatively stable. This makes me think that there was a specific group of patients (my guess would be head and neck cancer patients), who are particularly more vulnerable to pain, due to the modality of treatment employed (chemoradiation) which increases their pain as the treatment goes on, and pain improves only several weeks after the treatment is completed.

Response; None of the patients underwent any treatment other than chemotherapy during the follow-up period (3 treatment cycles).

Furthermore, the authors have not specified what the time intervals were between the moments 1, 2 and 3 in the pain evaluations.

Response: The three times were selected exclusively based on chemotherapy cycles (every two and three weeks, respectively) depending on the chemotherapy regimen used for the respective pathology.

Due to these concerns, I feel that the manuscript still needs significant changes before consideration for publication.

Potential suggestions for changes: include the 12 head and neck cancer patients and exclude the other patients. This will help compare scores and improvements in a uniform and homogenous population, making the results more meaningful. Exclude the patients with other cancers as their location and pathophysiology and treatment approaches are significantly different and an N of 3, 2 or 1 does not add much to the overall study.

Response: Your suggestion is welcome, but the number of patients would be even smaller for the study to be relevant if we excluded all patients with cancers other than head and neck cancers.

I suggest the authors to address if other palliative approaches such as radiation were utilized for pain relief in these patients.

Response: There were no other interventions except for pain treatment and chemotherapy.

I would recommend that the authors address the limitations of the study, including the small number of patients, lack of a control arm, single institutional study, etc.

Response: The limitations of the study have now been mentioned.  Now the text reads like:  “Limitations of the study: the small number of patients, lack of a control arm, single institutional study”.

Round 2

Reviewer 2 Report

Comments and Suggestions for Authors

All the comments are addressed.

Author Response

Rev 2: All the comments are addressed.

Thank you for taking the time to review the manuscript. As a complementary measure, we have grammatically corrected the text for the final version.

Reviewer 3 Report

Comments and Suggestions for Authors

The authors have acknowledged the limitations of the study. I suggest adding one additional limitation: heterogeneous group of patients with different malignancies and chemotherapeutic treatments were included in the study. 

Otherwise, the manuscript looks good, and okay to publish. 

Author Response

Rev 3: The authors have acknowledged the limitations of the study. I suggest adding one additional limitation: heterogeneous group of patients with different malignancies and chemotherapeutic treatments were included in the study. 

Response: At your recommendation, we have also introduced this limitation into the text, with which we fully agree.

Otherwise, the manuscript looks good, and okay to publish. 

Response: Thank you for taking the time to review the manuscript. As a complementary measure, we have grammatically corrected the text for the final version.